# Reduced non-Gaussianity by 30-second rapid update in convective-scale numerical weather prediction

Juan Ruiz[1,4], Guo-Yuan Lien[2], Keiichi Kondo[3], Shigenori Otsuka[4,5,6], and Takemasa Miyoshi[4,5,6,7,8]

[1]Centro de Investigaciones del Mar y la Atmósfera (CIMA-UBA/CONICET); Departamento de Ciencias de la Atmósfera y de los Océanos, FCEN, Universidad de Buenos Aires; Unidad Mixta Internacional-Instituto Franco-Argentino para el Estudio del Clima y sus Impactos (UMI-IFAECI/CNRS-CONICET-UBA), Buenos Aires, Argentina
[2]Central Weather Bureau, Taipei, Taiwan
[3]Meteorological Research Institute, Tsukuba, Japan
[4]RIKEN Center for Computational Science, Kobe, Japan
[5]RIKEN Cluster for Pioneering Research, Kobe, Japan
[6]RIKEN Interdisciplinary Theoretical and Mathematical Sciences Program, Kobe, Japan
[7]University of Maryland, College Park, Maryland, USA
[8]Japan Agency for Marine-Earth Science and Technology, Yokohama, Japan

**Correspondence:** Takemasa Miyoshi (takemasa.miyoshi@riken.jp), Juan Ruiz (jruiz@cima.fcen.uba.ar)

**Abstract.** Non-Gaussian forecast error is a challenge for ensemble-based data assimilation (DA), particularly for more non-linear convective dynamics. In this study, we investigate the degree of non-Gaussianity of forecast error distributions at 1-km resolution using a 1000-member ensemble Kalman filter, and how it is affected by the DA update frequency and observation number. Regional numerical weather prediction experiments are performed with the SCALE (Scalable Computing for Advanced Library and Environment) model and the LETKF (Local Ensemble Transform Kalman Filter) assimilating every-30-second phased array radar observations. The results show that non-Gaussianity develops rapidly within convective clouds and is sensitive to the DA frequency and the number of assimilated observations. The non-Gaussianity is reduced by up to 40% when the assimilation window is shortened from 5 minutes to 30 seconds, particularly for vertical velocity and radar reflectivity.

## 1 Introduction

The Kalman filter (KF) is the minimum variance linear unbiased estimator of the state of a dynamical system. The Ensemble Kalman Filter (EnKF, Evensen, 2009; Houtekamer and Zhang, 2016) is a Monte Carlo extension to the KF suitable for nonlinear systems with a large number of variables, so that it became a viable choice for data assimilation (DA) in numerical weather prediction (NWP) and other geoscience applications. The EnKF is optimal in the sense of maximum likelihood estimation when the error distributions are Gaussian (Evensen, 2009), but it becomes sub-optimal when the observational and forecast error distributions depart from the Gaussian (Lei et al., 2010). Miyoshi et al. (2014); Miyoshi et al. (2015) and Kondo and Miyoshi (2018) investigated non-Gaussianity in forecast error distributions using a 10,240 member EnKF with global atmospheric models. They showed that large non-Gaussianity measured by the Kullback-Leibler divergence is found frequently in the tropics mainly due to abundance of deep moist convection and also in other active areas with a real-world NWP model at

relatively low 112-km resolution. In those experiments, temperature and moisture show generally more non-Gaussian distributions than winds. Recently the horizontal resolution of operational NWP systems reached the order of 1 km, fine enough to resolve convective phenomena explicitly. Obtaining appropriate initial conditions at such high resolution is a challenge (Sun et al., 2014). The EnKF has been successfully applied to mesoscale assimilation of radar and satellite data (e.g., Stensrud et al., 2013). However, previous studies (Jacques and Zawadzki, 2014; Kawabata and Ueno, 2020) revealed that the underlying assumptions such as linear error dynamics and Gaussian error distributions are much more questionable in mesoscale than in synoptic and larger scales.

Miyoshi et al. (2016a, b) developed a so-called Big Data Assimilation (BDA) system assimilating observations every 30 seconds at 100-m resolution, taking advantage of new-generation technologies like the phased array weather radar (PAWR) which provides observations at unprecedented high temporal and spatial resolution. With the BDA configuration under an idealized Observing System Simulation Experiment (OSSE) framework, Maejima and Miyoshi (2020) showed that every-1-minute DA cycles resulted in better analyses than every-15-minute cycles. However, the impact of the DA frequency upon the forecast error distribution has not been investigated in real-case convective scale NWP.

This study investigates how the DA frequencies affect non-Gaussianity using a 1000-member, 1-km-mesh EnKF. 1000 ensemble members would be useful to detect non-Gaussian forecast error distributions as suggested by Kondo and Miyoshi (2019). Necker et al. (2020a, b) performed similar experiments and investigated the covariance structure and the effect of sampling noise at the mesoscale in a heavy rainfall event over Germany. Although the previous research employed data assimilation with only conventional observations at a 3-hourly DA frequency, this study is fundamentally different in the convection-resolving rapid DA cycles with PAWR data as frequently as every 30 seconds. The high frequency data allows us to investigate the sources of non-Gaussian distributions at the kilometer scale in the presence of rapidly-evolving deep moist convection. The paper is organized as follows: Section 2 describes methodological aspects. Results are presented in Section 3, and Section 4 provides concluding remarks and discussion.

## 2  Methodology

We use observations from the PAWR at Osaka University, Suita, Japan (Yoshikawa et al., 2013, Fig. 1a, red cross). This PAWR provides a unique dataset suitable for this study with various assimilation frequencies up to every 10 seconds at the fastest. This study follows the case study of Miyoshi et al. (2016a) focusing on the period between 0400 and 0600 UTC July 13, 2013, when heavy rains produced flash floods in Kyoto. Individual convective cells moved from west to east within a quasi-stationary intense rainband (see Fig. 1b for a snapshot at 0530 UTC). For this period, full volume scans of the PAWR are available every 30 seconds with 98 elevation angles, azimuthal resolution of $1.2°$ , and range resolution of 100 meters up to a maximum range of 60 km (Fig. 1a, red circle). Unambiguous Doppler velocities are available in the range $-50$ to $50 \ ms^{-1}$. PAWR reflectivity data is quality-controlled following Ruiz et al. (2015). A simple quality control algorithm has also been applied to the Doppler velocity field to remove outliers.

In this study, the regional NWP model known as the Scalable Computing for Advanced Library and Environment model (SCALE, Nishizawa et al., 2015) is used, coupled with the local ensemble transform Kalman filter (LETKF, Hunt et al., 2007). Lien et al. (2017) and Honda et al. (2018) describe the SCALE-LETKF system in detail. The model configuration follows Lien et al. (2017) with a single-moment bulk microphysics scheme (Tomita, 2008), a level-2.5 boundary layer turbulence scheme (Nakanishi and Niino, 2004), the Model Simulation Radiation Transfer radiation scheme (Sekiguchi and Nakajima, 2008), and soil processes represented by a Beljaars-type soil model (Beljaars and Holtslag, 1991).

The SCALE-LETKF system is implemented over a single domain with horizontal resolution of 1 km, and a size of 180 km by 180 km (Fig. 1a). 50 vertical levels extend up to 18 km elevation with a variable grid spacing from 140 m to 790 m in a hybrid sigma-z terrain-following coordinate. A 1000-member ensemble is used to assimilate the observations. Kondo and Miyoshi (2019) showed significant sampling error contaminations in non-Gaussian measures when the ensemble size is smaller than 1000. The initial conditions for the first cycle and the boundary conditions are taken from the National Centers for Environmental Prediction Global Data Assimilation System final analysis (FNL). Using FNL as the boundary conditions may be overly optimistic for the forecasting purpose, but this is not relevant to the goal of this study which focuses on non-Gaussian distributions and the impact of DA frequency.

The initial ensemble at the first assimilation cycle and the boundary condition ensemble are created by adding random perturbations which preserve the hydrostatic and nearly geostrophic equilibrium (Necker et al., 2020a; Maldonado et al., 2021). These perturbations are generated from a sample of continuous 6-hourly analysis states provided by the Climate Forecast System Reanalysis (CFSR, Saha et al., 2010), $\left[X_{CFSR}(t_1), X_{CFSR}(t_2), ...., X_{CFSR}(t_N)\right]$, where $N = 5840$ (4 years). The horizontal grid spacing of the CFSR data is $0.5°$. At the beginning of the assimilation cycle ($t = t_s$), the initial condition perturbation of the $i-th$ member $X'^{(i)}$ is computed as:

$$X'^{(i)}(t_s) = \alpha \left[X_{CFSR}(t_{n_1^{(i)}}) - X_{CFSR}(t_{n_2^{(i)}})\right]$$

where $\alpha$ is a multiplicative factor equal to 0.1 so that the amplitude of the perturbations is roughly equivalent to 10% of the climatological variability. The two CFSR analysis states are chosen by randomly selecting two numbers $n_1^{(i)}$ and $n_2^{(i)}$ from the N elements satisfying the condition that $t_{n_1^{(i)}}$ and $t_{n_2^{(i)}}$ correspond to the same time of the year and time of the day. In the following assimilation cycles at time $t > t_s$, we obtain the boundary perturbations as:

$$X'^{(i)}(t) = \alpha \left[(1-\beta)\left(X(t_{l_1^{(i)}}) - X(t_{l_2^{(i)}})\right) + \beta\left(X(t_{u_1^{(i)}}) - X(t_{u_1^{(i)}})\right)\right]$$

where $l_{1,2}^{(i)} = n_{1,2}^{(i)} + m$ and $u_{1,2}^{(i)} = n_{1,2}^{(i)} + m + 1$, with $m = floor[(t-t_s)/6h]$ and $\beta = [(t-t_s)/6h] - m$ being a temporal linear interpolation factor to compute perturbations at arbitrary times (not necessarily a multiple of 6h). In this way we obtain perturbations that are smoothly varying in time and consistently with the large scale dynamics of the atmosphere. This procedure is applied to all atmospheric and soil state variables.

In the SCALE-LETKF system, radar data can be assimilated using different localization scales for different variables. Based on preliminary experiments with the SCALE-LETKF using smaller ensemble sizes and every-30-second PAWR data, it was

found that a vertical localization scale of $2km$ (with a $7.3km$ cut-off, similarly hereafter) produced good results. For horizontal

localization, better results were obtained using $4 - km$ localization to assimilate observations with reflectivities $> 10dBZ$. Observations of reflectivity values $\leq 10dBZ$ are assimilated with a fixed value of $10dBZ$ to avoid large observation-minus-forecast departures associated with clear air reflectivities (Aksoy et al., 2009). Also, a shorter horizontal localization scale of $2km$ is used to reduce the impact of non-precipitating observations at the edge of clouds. Doppler velocity observations are assimilated with horizontal and vertical localization scales of $10km$ and $3km$, respectively. For covariance inflation, a

relaxation to prior ensemble spread (RTPS, Whitaker and Hamill, 2012) with a relaxation parameter of $0.9$ is applied. This helps consider the inhomogeneous distribution of observations as in Lien et al. (2017).

Reflectivity and Doppler velocity observations are superobbed to horizontal resolution of $1km$ and vertical resolution of $500m$ to approximately match the model resolution. This helps reduce the errors of representativeness due to the gap between what is represented by the model and observation. This procedure can also reduce the impact of possible spatial correlations in

the observation errors. The observational error standard deviations for these super-observations are set at $5.0dBZ$ and $3.0ms^{-1}$ for reflectivity and Doppler velocities, respectively. The radar data are assimilated up to a maximum height of $11km$.

A spin-up DA experiment with every-5-minute PAWR reflectivity and Doppler velocity data is performed for an hour from 0400 UTC, July 13, 2013. Only a single PAWR volume scan closest to the analysis time is assimilated per analysis. The 1000-member analysis ensemble at 0500 UTC is used as the initial conditions for the DA experiments.

Experiments are performed with different DA update frequencies to study the impacts of the DA frequency and observation number on the forecast error distributions. All experiments share the configuration described above, but the only differences are the DA frequency and the amount of the data assimilated. First, four experiments with 5, 2, 1, and 0.5 minutes DA frequencies are performed, hereafter referred to as 5MIN, 2MIN, 1MIN, and 30SEC, respectively. Here, only a single volume scan closest to the analysis time is used per analysis. Namely, more frequent updates assimilate more data. In all cases the time difference

between the observation time (center time of the radar volume scan) and the analysis time do not differ by more than 15 seconds.

Next, to separate the impact of DA frequency and the amount of data assimilated, two additional experiments are performed using a 5-minute and 1-minute DA frequency, with all radar volumes every 30 seconds assimilated by a 4-dimensional EnKF approach Hunt et al. (2004). These experiments are referred to as 5MIN-4D and 1MIN-4D, respectively, assimilating the same

amount of data as 30SEC but using longer assimilation windows.

To measure the degree of non-Gaussianity of the error distributions we compute the Kullback-Leibler divergence (hereafter KLD, Kullback and Leibler, 1951) which is defined as follows:

$$KLD(P\|Q) = \int\limits_{-\infty}^{\infty} p(x) \ln \frac{p(x)}{q(x)} dx, \qquad (1)$$

where $p(x)$ and $q(x)$ are the probability density functions (PDFs) of P and Q, respectively. The KLD is 0 if $P$ and $Q$ are the

same and takes positive values if $P$ and $Q$ differ. In our case $p(x)$ is either the first guess or analysis error distribution for the state variable $x$, and $q(x)$ is a Gaussian distribution whose mean and standard deviation are equal to the ones of $p(x)$.

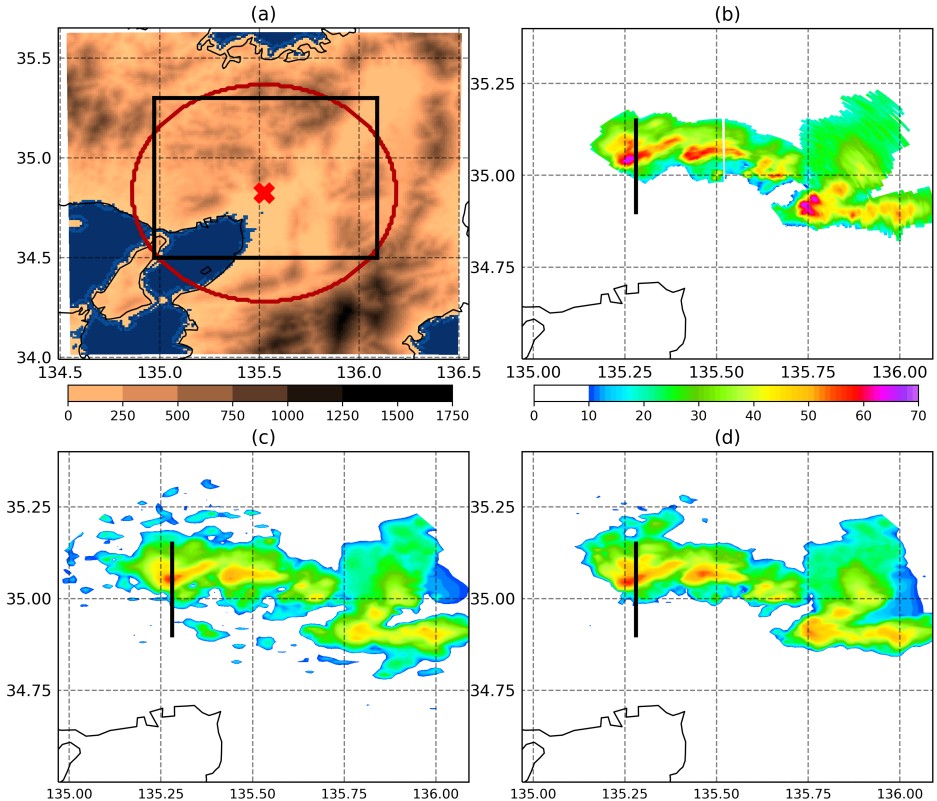

**Figure 1.** (a) Terrain height of the 1-km-mesh SCALE-LETKF domain (shades, m). The red circle indicates the 60-km radar range centered at the radar site (red cross) in Osaka University, Suita, Japan. The black box indicates the area shown in (b-d). (b) column-maximum PAWR observation (dBZ) at 0530 UTC, half an hour after the initialization of the data assimilation cycle, (c) 5MIN and (d) 30SEC experiments analysis ensemble mean column-maximum radar reflectivity (dBZ) at 0530 UTC. Black lines indicate the locations of the cross-sections displayed in Fig. 2

Therefore, a low KLD value corresponds to the first guess or analysis error distribution close to a Gaussian. In the EnKF we do not have access to the continuous PDF $p(x)$ but to its finite, limited sample. For each state variable $x$ (e.g. temperature, wind components, etc) and at each model grid point, we approximate $p(x)$ with the sample histogram from the 1000-member
ensemble using 32 equally-sized bins covering the range where $p(x)$ is greater than 0. This range is defined by the minimum and maximum values of $x$ at each model grid point and time. Hence, we can approximate the KLD as follows:

$$KLD(P\|Q) \approx \sum_{j=1}^{j=32} p_j \ln \frac{p_j}{q_j}, \tag{2}$$

where $p_j$ is the empirical frequency of $x$ at the $j-th$ histogram bin. $q_j$ is the integral over the $j-th$ histogram bin of a Gaussian PDF whose mean and standard deviation are given by the ensemble-based sample estimates. After implementing this

we end up with an estimation of the KLD of the analysis and first guess error distributions with respect to the Gaussian for each grid point location, vertical level, and time.

## 3 Results

All experiments show that the analyzed reflectivity fields are in good agreement with the observation. However, some differences can be found between the experiments that assimilate different amounts of data and with different assimilation windows.

For example, Figures 1c and d show that 30SEC captures the strong reflectivity areas (>45 dBZ, orange and red shadings) better than 5MIN. 5MIN shows noisy patterns of spurious convective cells surrounding the main convective rainband.

### 3.1 Impact of DA frequency on the analysis of a convective cell

In this section we explore the impact of data assimilation frequency over a convective cell located in the cross section along the black line in Fig. 1b-d at 0530UTC. First, the impact of data assimilation frequency is explored by the 5MIN, 2MIN, 1MIN,

and 30SEC experiments. Here, more observations are assimilated with more frequent data assimilation. Figure 2 top row (a-d) shows that the reflectivity patterns (Z, shades) are similar among all experiments, but vertical velocity (W, contours) are different. Stronger updrafts are found in DA experiments with shorter assimilation windows. This suggests that DA frequency have a significant impact upon quantities which are not directly observed.

Strong non-Gaussianity is observed in the first-guess ensemble in W and temperature T in the 5MIN experiment (Figure 2e

and i respectively). Non-Gaussianity for W is stronger at the southern edge and the highest peak of the convective cell, which is probably related to the development of a new updraft in the southern edge and the top of the strong updraft, respectively. Weaker low-level maxima south of the convective line are associated with shallow convective clouds that are not effectively corrected by radar observations. The KLD maxima for T are approximately collocated as those for W. KLD maxima in T can be associated with non-Gaussianity in W through vertical advection of scalar quantities such as T and moisture. Another KLD

maximum for T is found near the surface south of the convective cell, probably associated with the gust front.

Kondo and Miyoshi (2019) found that in synoptic scales, the ensemble spread maxima are collocated with the KLD maxima. At convective scales for W, the ensemble spread maxima (Fig. 2e, red contours) are not necessarily collocated. For example, larger departures from the Gaussian are found above the ensemble spread maximum associated with the main updraft in the 5MIN experiment. For temperature also there is no clear relation in the distribution of the ensemble spread and the KLD,

although KLD maxima seem to occur within areas of relatively large ensemble spread. As the assimilation frequency increases it is more difficult to find a relationship between KLD and ensemble spread either for W or T (Fig. 2 second and third rows).

KLD for W and T are consistently reduced with more frequent DA (Fig. 2e-h), although the reduction is smaller for T. Overall, KLD is reduced more from 5MIN to 2MIN than from 1MIN to 30SEC. This reduction occurs mainly within the convective clouds. Non-Gaussianity in W at low levels observed outside the cloud is not significantly affected by more frequent

updates. The ensemble spread for W is also reduced with more frequent DA and indicates a narrower error distribution. This

result is linked with the reduced non-Gaussianity since it is expected that smaller perturbations grow in a more linear regime and contribute to reducing the departures from the Gaussian.

To better investigate the shape of the error distributions and how they are affected by the update frequency, Fig. 3 shows the sample histograms for the first guess at the location of maximum KLD (indicated with a black cross in Fig. 2). We restrict the search of the maximum KLD to the grid points at which the ensemble mean reflectivity is over 30 dBZ where radar data impact would be large. The forecast error distribution for W and for the 5MIN experiment shows large departures from the Gaussian with a strong positive tail (Fig. 3a). A similar situation is observed for Z (Fig. 3e). This result is consistent since ensemble members with larger W are probably associated with larger reflectivity values, so both distributions become positively skewed. As the update frequency is increased, non-Gaussianity and ensemble spread are reduced for both variables. The only exception is Z at 30SEC update frequency that shows a KLD value that is slightly larger than that in the 1MIN experiment. Note that these error distributions are taken at slightly different locations based on the simulated convection locations in each experiment and thus the mean of the distribution can change from one experiment to the other.

4D-EnKF experiments allow us to investigate the impact of changing the assimilation frequency while keeping the observation number unchanged. 5MIN-4D shows weaker updrafts (similar to those found in 5MIN) compared with experiments with more frequent updates (Fig. 4a,b). 5MIN-4D also shows almost the same ensemble spread for W and T as 5MIN (Fig. 4c and e, red contours). KLD for W (Fig. 4c, shades) is lower, indicating that the observation number contributes to reducing non-Gaussianity. This is not the case for T for which KLD is similar or larger (Fig. 4e, shades). 1MIN-4D is close to 1MIN and 30SEC in terms of non-Gaussianity and the shape and strength of the convective cell (Figs. 4b, d and f).

## 3.2 Spatio-temporal distribution of non-Gaussianity

We further investigate the non-Gaussianity by averaging the KLD vertically and temporally (Fig. 5). In 5MIN, the central and eastern sides of the convective area show relatively low KLD values because the impact of radar DA is generally bigger in the convective areas (Fig. 5a). The impact of DA frequency on non-Gaussianity is investigated by means of the relative KLD difference between the 5MIN and all the other experiments, computed as:

$$\overline{KLD_{diff}} = \frac{\overline{KLD_E} - \overline{KLD_{5MIN}}}{\overline{KLD_{5MIN}}}, \qquad (3)$$

where $\overline{KLD_{diff}}$ is the relative difference between the averaged KLD in the 5MIN experiment ($\overline{KLD_{5MIN}}$) and on each of the other experiments ($\overline{KLD_E}$), where $E$ can be either 5MIN-4D, 2MIN, 1MIN, 1MIN-4D or 30SEC).

KLD consistently decreases with increasing DA frequency (Figs. 5b-d). KLD is reduced by up to 40% in 30SEC with respect to the 5MIN. KLD is reduced more in the convective area, where more observations are assimilated. Increasing the DA frequency and the observation number produces a more substantial impact over the western part of the convective line where KLD maxima are found associated with convective cells entering the radar range from the west.

KLD in 1MIN-4D is as low as that in 30SEC and lower than that in 1MIN. This result suggests that both observation number and DA frequency contribute to reducing non-Gaussianity, at least for high DA frequencies. KLD in 5MIN-4D is lower than

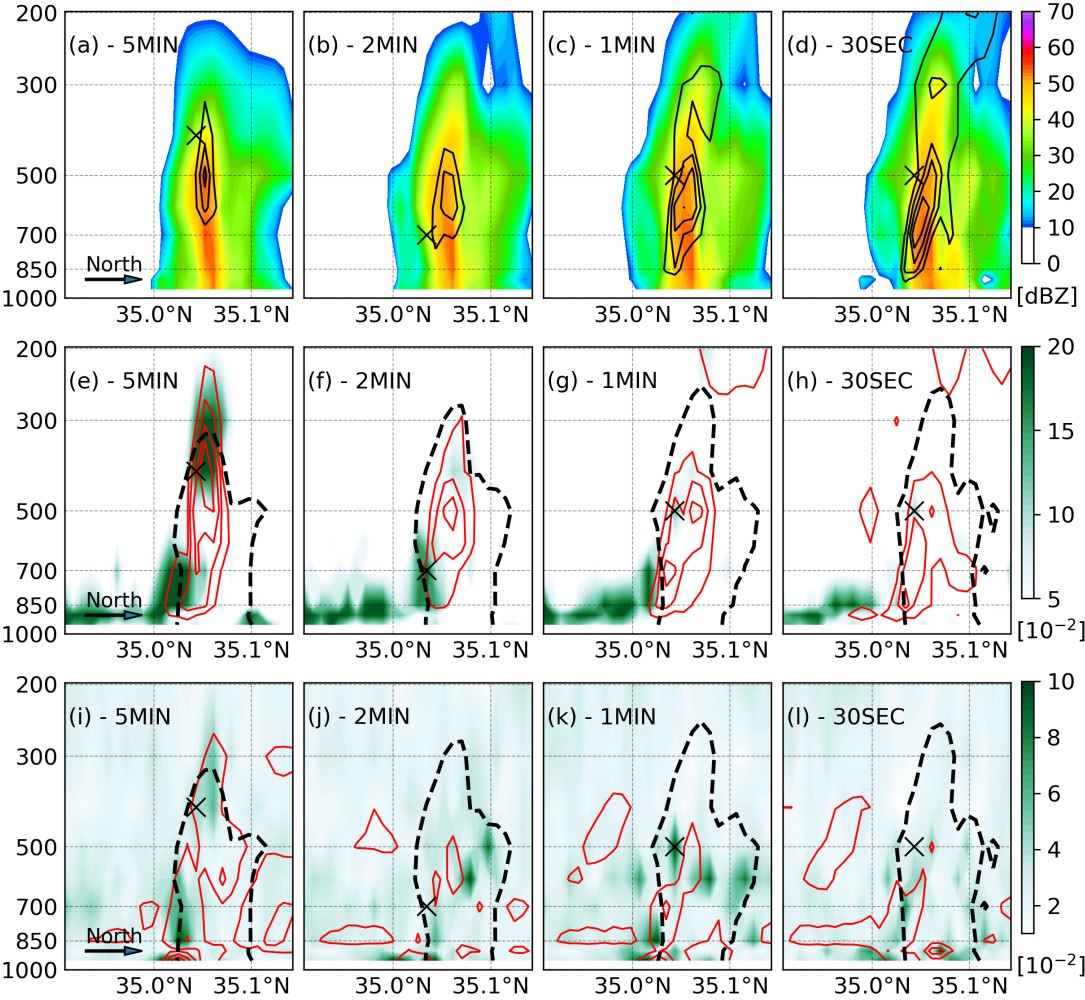

**Figure 2.** (a-h) South-North vertical cross-section along the black line indicated in Fig. 1b-d at 0530 UTC for (a-d) first-guess ensemble-mean reflectivity (Z, shades, dBZ) and vertical velocity (W, contours every 2.5 $ms^{-1}$), (e-h) vertical velocity KLD (shades, $10^{-2}$) and ensemble spread (red contours at 1.0, 2.5, 5.0 and 10.0 $ms^{-1}$), and (i-l) temperature KLD (shades, $10^{-2}$) and ensemble spread (red contours at 0.2, 0.5, 1.0 and 2.0 $K$). Blacked dashed contours indicate reflectivity over 30 dBZ. The black cross in panels (i-l) indicates the location of the maximum KLD within the grid points at which $Z > 30 dBZ$.

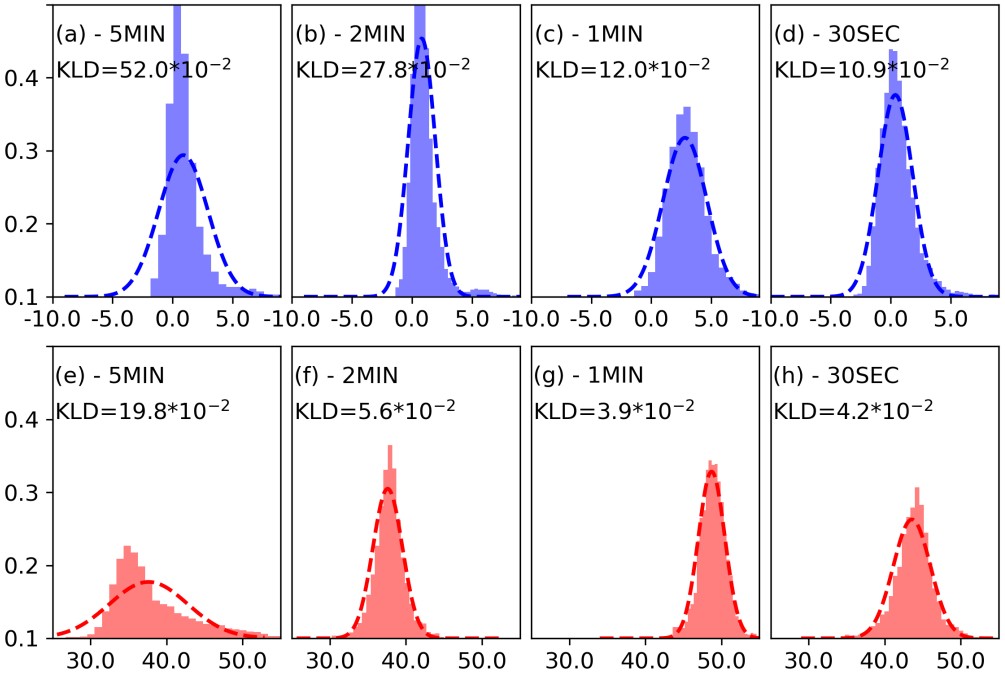

**Figure 3.** Sample histograms for (a-d) vertical velocity ($ms^{-1}$) and (e-h) reflectivity (dBZ) at the location of the maximum KLD for vertical velocity (black cross in Fig. 2). Thick dashed curves indicate fitted Gaussian functions, and KLD non-Gaussian measures are also indicated. Each column corresponds to 5MIN, 2MIN, 1MIN, and 30SEC from left to right, respectively.

that in 5MIN, so that a larger observation number helps to reduce non-Gaussianity. However, KLD in 5MIN-4D is larger than that in 30-SEC or 1MIN-4D, indicating that DA frequency is equally important. Moreover, the impact of DA frequency can be larger in the case of variables like T and moisture. As already found in the vertical cross-sections (Fig. 4), for those variables, KLD in 5MIN and 5MIN-4D is almost the same, while KLD is clearly reduced for 1MIN, 1MIN-4D, and 30SEC (not shown).

We also investigate the vertical distribution of non-Gaussianity by the spatially averaged vertical profile of KLD at "precipitating" grid points, defined by the ensemble-mean column-maximum reflectivity $> 30dBZ$, and "non-precipitating" grid points, defined by the ensemble-mean column-maximum reflectivity $< 0dBZ$. At the "precipitating" grid points (Figures 6a-d) KLD for temperature and vertical velocity is maximum at mid-levels coinciding with the maximum in latent heat release within convective clouds and with the maximum ensemble spread for these two variables (not shown). KLD for temperature, vertical velocity, and specific humidity maximizes at lower heights over the non-precipitating area since, as stated before, at such locations non-Gaussianity is mainly associated with shallow convection. For instance, for the vertical velocity, the ensemble spread in the shallow convection is usually low, but the KLD can be larger. An upper-level maximum in KLD is found for the meridional wind (Figs. 6d and h), also coinciding with the maximum ensemble spread (not shown). Convective outflows are stronger at the top of convective clouds and can be one of the mechanisms contributing to the increase of non-Gaussianity at these levels over the precipitating area. Overall, KLD in 30SEC is lower than that in 5MIN with reductions of more than 40%.

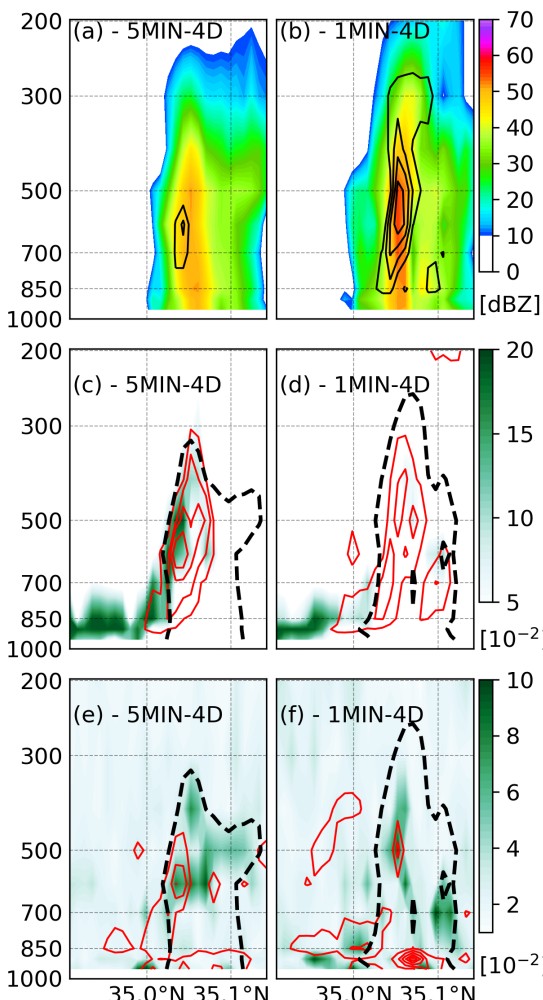

**Figure 4.** As in Fig. 2 but for (a, c and e) 5MIN-4D and (b, d and f) 1MIN-4D.

The reduction of KLD in the non-precipitating area is smaller because the radar DA is inherently less effective in these areas (Figs. 6e-h). There are some exceptions to the general reduction in non-Gaussianity with increased update frequency. Specific humidity in no-precipitating grid points shows larger KLD in the 30SEC than in the 5MIN experiments. This is also the case for the precipitating grid points at upper levels in the second half of the experiment. Also the KLD in W in the no-precipitating grid points at middle and upper levels is slightly larger in the 30SEC experiment.

### 3.3 Non-Gaussianity evolution within the DA cycle

To investigate the effect of the analysis update on non-Gaussianity we present the time series of the KLD of the analysis and first guess vertically and horizontally averaged over the "precipitating" and "non-precipitating" grid points (Fig. 7). At

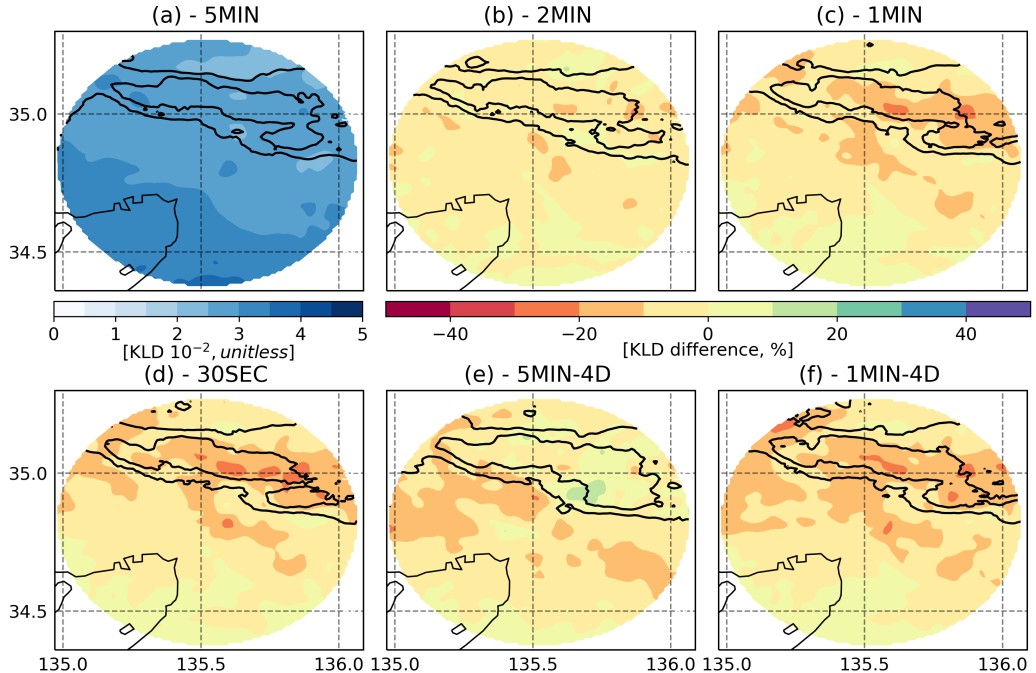

**Figure 5.** (a) Column-averaged KLD for zonal wind for 5MIN, averaged for the experiment period from 0500 to 0600 UTC. Relative KLD difference (%) from 5MIN for (b) 2MIN, (c) 1MIN, (d) 30SEC, (e) 5MIN-4D, and (f) 1MIN-4D. Warm colors correspond to smaller KLD values.

most times and variables over the "precipitating" and "non-precipitating" grid points, KLD is reduced during the assimilation step. Experiments with longer windows show more KLD growth during the forecast as expected, but also a larger reduction at the analysis step, which is not as effective as the more frequent updates in reducing the analysis KLD. As noted before, the specific humidity over the "non-precipitating" grid points behaves differently, and KLD increases during the assimilation

215    step for almost all times and experiments, leading to larger KLD at shorter assimilation windows (Figs. 6b and f). In this area mostly "non-precipitating" observations are assimilated to suppress spurious clouds. Interestingly in the "non-precipitating" grid points 5MIN-4D is the experiment providing the lowest KLD for all variables (Figs. 7b, d and f). This result suggests the potential benefits of treating "non-precipitating" observations differently.

      To evaluate the impact of assimilation frequency on the distance between the analysis and first guess to the observations in

220    a more systematic way, we compute the root mean squared error (RMSE) and bias for reflectivity observations (Fig. 8). The computation of the RMSE and bias between the model and the observations is done by comparing the column maximum of the reflectivity for each horizontal grid location and time. The RMSE and bias are computed only over grid points at which the observed maximum reflectivity is over $5dBZ$. The time series of RMSE shows a better fit to the observed reflectivity for shorter assimilation windows. The impact of 4D DA is not so clear, 1MIN-4D slightly outperforms the 1MIN but 5MIN-4D and

225    5MIN perform similarly (Fig. 8a). This is partially because in 4D data assimilation the analysis results from the assimilation

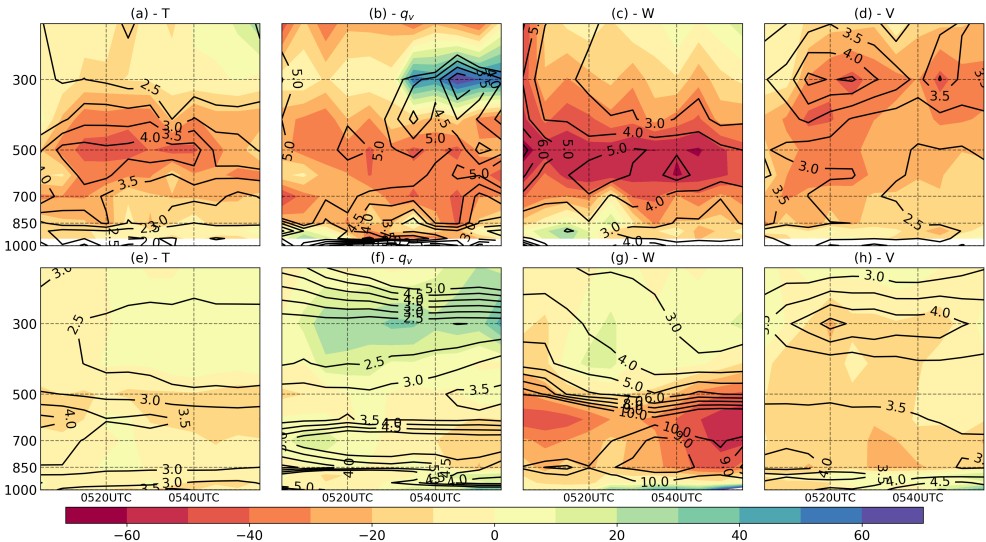

**Figure 6.** Time-vertical cross-section of KLD for 5MIN (contours, $10^{-2}$) and the relative difference for 30SEC (shaded), averaged over the (a-d) "precipitating" (>30dBZ) and (e-h) "non-precipitating" (<0dBZ) grid points for (a,e) temperature, (b,f) specific humidity, (c,g) vertical velocity and (d,h) meridional wind.

of all the observations within the assimilation window, while to construct this figure, only the observations at the analysis time were considered. The bias, computed as the mean difference between the model and the observations does not seem to be consistently affected by the assimilation frequency (Fig. 8b). These results are in agreement with those observed in the time series of KLD for different variables. However, we should be cautious with the interpretation of these results since increasing the observation number can lead to both a reduced KLD and a better fit to the observed quantities, not necessarily implying a causal link between these two effects.

## 4 Summary and Discussion

One-thousand-member 1-km-resolution ensemble DA experiments were performed using real phased array radar observations and a mesoscale NWP model to investigate the impact of DA frequency and observation number on the non-Gaussian error distributions. We found that a DA frequency of 5 minutes, although it was already much faster than the typical DA frequency, resulted in strong non-Gaussianity possibly affecting the performance of the EnKF. Non-Gaussianity is stronger for vertical velocity as has been found by Kawabata and Ueno (2020). Non-Gaussianity is also larger at mid-levels within convective cells, near the level of larger latent-heat release and vertical accelerations associated with convective instability. At convective scales, some of the local maxima in KLD can be related directly to advection by mesoscale circulations associated with strong convective cells, but other processes not specifically presented in this study may also possibly contribute to the generation of non-Gaussianity, such as those not directly associated with clouds, like differential heating circulations or gravity waves.

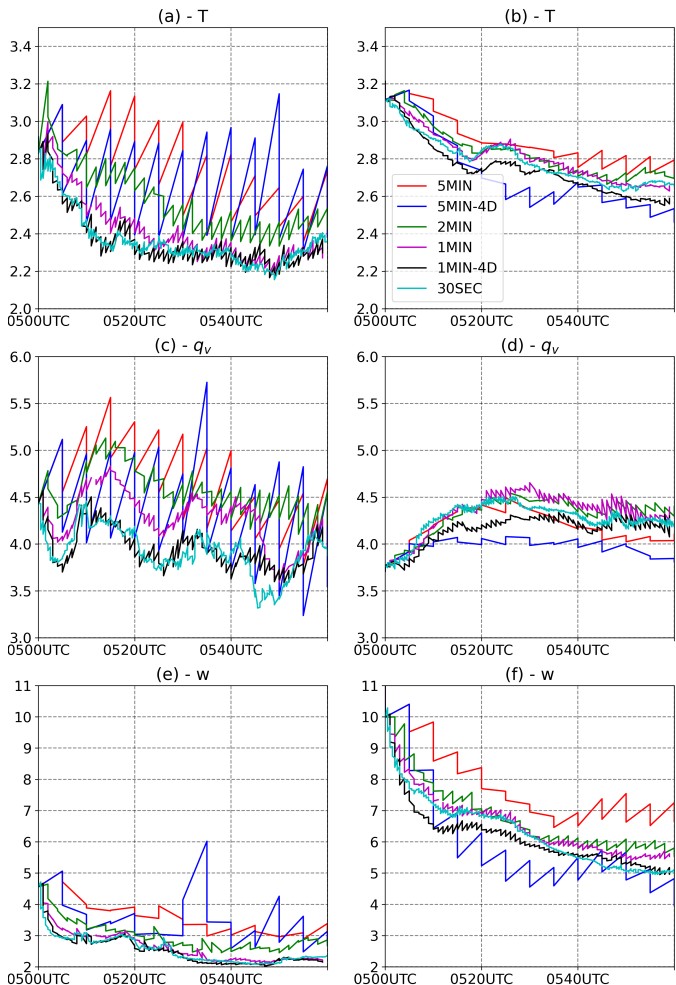

**Figure 7.** Sawtooth time-series of the KLD ($10^{-2}$) of the analysis and first guess vertically and horizontally averaged over the precipitating (>30dBZ, a,c,e) and non-precipitating (<0dBZ, b,d,f) grid points for temperature (a,b), specific humidity (c,d) and vertical velocity (e,f) and for the 5MIN (red), 5MIN-4D (blue), 2MIN (green), 1MIN (magenta), 1MIN-4D (black) and 30SEC (cyan) experiments.

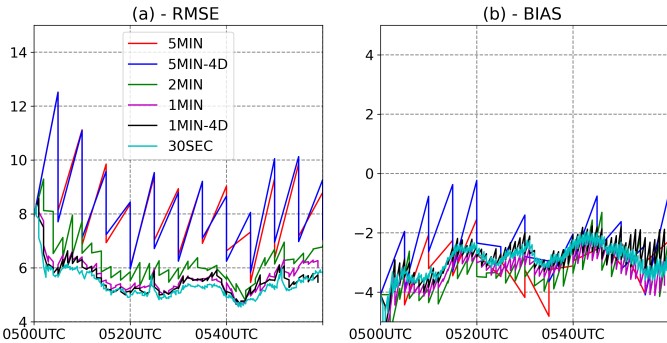

**Figure 8.** Sawtooth time-series of the root mean squared error (dBZ, a) and bias (dBZ, b) of the maximum reflectivity of the analysis and first guess for the 5MIN (red), 5MIN-4D (blue), 2MIN (green), 1MIN (magenta), 1MIN-4D (black) and 30SEC (cyan) experiments.

We found that increasing the analysis update frequency and observation number from 5 minutes to 30 seconds has a huge impact upon non-Gaussianity in the error distributions for all model variables but particularly for vertical velocity and reflectivity which are the ones showing larger KLD from Gaussianity at these scales. Increasing the assimilation frequency to 30 seconds and assimilating more observations can reduce KLD by up to $40\%$. Moreover, 4D-EnKF experiments revealed that for frequent DA of every 1 minute, the observation number explained most of the reduction in non-Gaussianity; in contrast, for a longer window of 5 minutes, even the experiments using all 30-second-frequency observations presents significant departures from the Gaussian. While convective clouds are particularly favorable for nonlinear error growth, non-Gaussianity is not necessarily larger within convective clouds. This is mainly due to the convective-scale radar DA is usually most effective within precipitating clouds.

There are two possible ways in which more frequent DA can result in error distributions closer to the Gaussian. First, more frequent DA contributes to a quasi-linear evolution of the forecast error due to forecast lengths which are shorter than the predictability limit for the resolved scales. This also helps keep the perturbation small and can additionally contribute to quasi-linear perturbation dynamics. Second, our results show that the analysis step effectively contributes to reducing non-Gaussianity for different variables, although this may not be the case for "non-precipitating" reflectivity observations that produce an increase in KLD for specific humidity. Non-gaussianity reduction during DA is larger with longer windows. However, it is not enough to compensate for the effect of more rapid and non-linear error growth during the forecast step in the lower update frequency experiments.

From the point of view of KLD reduction, the largest impact is found between 5MIN and 2MIN updates. This suggests that non-linear error growth become more important after the first 2 minutes of integration at these scales. This hypothesis is partially supported by the reduction in RMSE and ensemble spread. A 2-minute update frequency seems to provide a good compromise between the computational cost and non-Gaussianity of the error distributions. However, from the point of view of the analysis accuracy more frequent DA provides a better fit to the observed quantities. The specific role of reduced non-Gaussianity on this is not clear and should be further investigated. Gaussian error distributions may contribute to more accurate

analysis updates, but in the current experimental setting, other factors like the increase in the number of assimilated observations may also lead to the reduction in the RMSE for observed quantities. Maejima and Miyoshi (2020) investigated the impact of assimilation frequency at 1-km using observing system simulation experiments. They also found a significant improvement in the forecast quality when the assimilation window is reduced from 5 minutes to 3 minutes and additional improvements using 1 minute windows. These results are consistent with what is found in this paper with respect to Gaussianity in the error
distributions.

Moreover, as has been shown in the previous studies, more frequent assimilation can produce a larger degree of imbalance in the initial conditions which can degrade the quality of the forecasts (e.g., Lange and Craig, 2014; Bick et al., 2016). Therefore, despite the potential benefits of a more Gaussian model error distribution on the analysis accuracy, other factors may degrade the forecasts initialized from more frequent data assimilation cycles. Imbalance may also be an additional source
of non-Gaussianity. Gaussian error distributions can lead to more physically meaningful assimilation updates in the context of an EnKF and therefore, more balanced initial conditions. However, a larger imbalance in the initial conditions can contribute to faster error growth and increased departure from the Gaussian in the forecast distribution. Possible interactions of these mechanisms in a data assimilation cycle have not been investigated, and are a subject for future research. Our results suggest that despite the effect of a larger imbalance, the increase of DA frequency reduce non-Gaussianity in the sample distributions
with the EnKF. This is even true to variables like vertical velocity within convective clouds which are frequently used to measure the effect of imbalance in the initial conditions.

This study is the first attempt to investigate the impact of assimilation frequency and observation number on non-Gaussianity using an EnKF employing a large 1000-member ensemble and every-30-second observations from a PAWR. In this first set of experiments, we evaluate the impact on the non-Gaussianity of the ensemble-based sample distribution. Future experiments
will be performed to investigate the overall quality of the analysis obtained with different assimilation windows and number of observations and also the impact of assimilation window upon the structure of the error covariance matrix.

*Code and data availability.* The codes used for the main results of this study can be accessed at a public github repository (https://github.com/takemasa-miyoshi/letkf). Essential data to reproduce the results of this study are stored for 5 years in RIKEN R-CCS. Due to the large volume of data and limited disk space, data will be shared online upon request (takemasa.miyoshi@riken.jp). The phased array radar data can be visualized
at https://pawr.nict.go.jp/index_en.html

*Author contributions.* All the authors participate in the conception of the ideas, in the design of the experiments and in draft manuscript preparation. Guo-Yuan Lien wrote the SCALE-LETKF code used in this study. Juan Ruiz conducted the experiments and analyzed the results. All authors reviewed the results and approved the final version of the manuscript.

*Competing interests.* The authors declare that they have no conflict of interest.

*Acknowledgements.* We thank the members of RIKEN CCS Data Assimilation Research Team and JST AIP project for valuable discussions. The PAWR data was provided by the National Institute of Information and Communications Technology (NICT) science cloud system. This work was supported by CREST, JST projects 'EBD: Extreme Big Data – Convergence of Big Data and HPC for Yottabyte Processing' (grant number: JPMJCR1303), 'Innovating "Big Data Assimilation" technology for revolutionizing very-short-range severe weather prediction' (grant number: JPMJCR1312), JST AIP acceleration research 'Big Data Assimilation and AI Creating New Development in Real-time

Weather Prediction' (grant number JPMJCR19U2) and Advancement of meteorological and global environmental predictions utilizing observational "Big Data" of the social and scientific priority issues (Theme 4) to be tackled by using FUGAKU computer of the FLAGSHIP2020 Project of the Ministry of Education, Culture, Sports. This research used computational resources of the K-computer at RIKEN R-CCS (Project IDs ra000015, ra001011, hp150019, hp160162, hp170178, hp180062, hp190051), Post-K project ID hp 120282 and the Joint Center for Advanced High Performance Computing (JCAHPC) Oakforest-PACS (Project IDs hp190051 and hp200026). This study was also sup-

ported by grants PICT-2033/2017 for Agencia Nacional de Promoción Científica y Tecnológica and 20020170100504 BA from the University of Buenos Aires, and by JSPS KAKENHI Grant Number JP16K17807.

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
