# Peer review of "Reduced non-Gaussianity by 30-second rapid update in convective-scale numerical weather prediction"

_Nonlinear Processes in Geophysics, 2021_

## Author Comment (AC1)

**Reviewer #1**

Review of "Reduced non-Gaussianity by 30-second rapid update in convective-scale numerical weather prediction " by J. Ruiz et al..

General comments: The manuscript investigated the degree of non-Gaussianity of forecast error distributions and how it is affected by the DA update frequency and observation number. This article used 1000 ensembles, but the generation of ensembles is not very clear. The introduction of the KLD method is not clear. Some words are not rigorous. This paper theoretically provides evidence that increasing the update frequency and the observations can improve the accuracy of assimilation in convective-scale. I think this manuscript can be considered for publication if these concerns could be addressed:

We would like to thank the reviewer for the comments which help to improve the clarity of the presentation and also bring additional discussions to the paper. Based on the reviewer's comments we expand the description of the methodology and add additional explanations and discussions.

Specific comments:

1. The generation of ensembles need to be described in detail, because it is very important for this article and the ensemble DA method.

We agree that this is an important aspect of the experimental design so following the reviewer's comment we expand the description on how the initial perturbations for the first assimilation cycle as well as the boundary conditions are generated in our experiments.

The initial ensemble at the first assimilation cycle and the ensemble boundary condition ensemble are created by adding random perturbations which preserve the hydrostatic and nearly geostrophic equilibrium (Necker et al. 2020a, Maldonado et al. 2021). These perturbations are generated from a sample of continuous 6-hourly analysis states provided by the Climate Forecast System Reanalysis (CFSR, Saha et al. 20202), $[X_{CFSR}(t_1), X_{CFSR}(t_1),...,X_{CFSR}(t_N)]$, where N=5840 (4 years). The horizontal grid spacing of the CFSR data is 0.5º. At the beginning of the assimilation cycle (t=$t_s$), the initial condition perturbation of the *i-th* member $X'^{(i)}$ is computed as:

$$X'^{(i)}(t_s) = \alpha \left[ X_{CFSR}(t_{n_1}^i) - X_{CFSR}(t_{n_2}^i) \right]$$

where $\alpha$ is a multiplicative factor equal to 0.1 so that the amplitude of the perturbations is roughly equivalent to 10% of the climatological variability. The two CFSR analysis states are chosen by randomly selecting two numbers $n_1^{(i)}$ and $n_2^{(i)}$ from the N elements satisfying the condition that $t_{n1(i)}$ and $t_{n2(i)}$ correspond to the same time of the year and time of the day. In the following assimilation cycles at time t>$t_s$, we obtain the boundary perturbations as:

$$X''^{i}(t) = \alpha \left[ (1-\beta) \left( X(t_{l_1^{(i)}}) - X(t_{l_2^{(i)}}) \right) + \beta \left( X(t_{u_1^{(i)}}) - X(t_{u_1^{(i)}}) \right) \right]$$

where $l^{(i)}_{1,2} = n_{1,2}^{(i)} + m$ and $u^{(i)}_{1,2} = n_{1,2}^{(i)} + m + 1$, with $m$ = floor$[(t - t_s)/6\ h]$ and $\beta = [(t-t_s)/6\ h] - m$ being a temporal linear interpolation factor to compute perturbations at arbitrary times (not necessarily a multiple of 6 h). In this way we obtain perturbations that are smoothly varying in time and consistently with the large scale dynamics of the atmosphere. This procedure is applied to all atmospheric and soil state variables.

2. The introduction of KLD should contain how to operate in this article? What should be noted? So that readers can repeat your experiment

We agree with the reviewer and we expanded the description on how KLD is computed in this work. In particular, we added more details on how KLD is computed from the ensemble-based sample, and an equation expressing how KLD is computed in this paper.

To measure the degree of non-Gaussianity of the error distributions we compute the Kullback-Leibler divergence (KLD, Kullback 1951) which is defined as follows:

$$KLD(P\|Q) = \int_{-\infty}^{\infty} p(x) \ln \frac{p(x)}{q(x)} dx,$$

where $p(x)$ and $q(x)$ are the probability density functions (PDFs) of P and Q respectively. The KLD is 0 if P and Q are the same and takes positive values if P and Q differ. In our case $p(x)$ is either the first guess or analysis error distribution for the state variable x (e.g. temperature, wind components, etc), and $q(x)$ is a Gaussian distribution whose mean and standard deviation are equal to the ones of $p(x)$. Therefore, a low KLD value corresponds to the first guess or analysis error distribution close to a Gaussian.
In the EnKF we do not have access to the continuous PDF $p(x)$ but to a finite sample of it. For each state variable x and at each model grid point, we approximate $p(x)$ with the sample histogram populated from the 1000-member ensemble using 32 equally-sized bins covering the range where $p(x)$ is greater than 0. This range is defined by the minimum and maximum values of x at each model grid point and time. Then we can approximate the KLD as follows:

$$KLD(P\|Q) \approx \sum_{j=1}^{j=32} p_j \ln \frac{p_j}{q_j},$$

where $p_j$ is the relative frequency of x at the *j-th* histogram bin. $q_i$ is the integral over the *j-th* histogram bin of a Gaussian PDF whose mean and standard deviation are given by the ensemble-based sample estimates. After implementing this we end up with an estimation of

the KLD of the analysis and first guess error distributions with respect to the Gaussian for each grid point location, vertical level, and time.

3. Is the solution of KLD grid point by grid ? Using the assimilated ensembles to statistics?

Following this comment and the previous one we extend the description on how KLD is computed so that the experiments can be reproduced. The specifications on how KLD is computed from the ensemble members is included (see also the answer corresponding to the previous comment).

4. Please give the formula of relative KLD difference.

We include an equation for the relative KLD difference.

The impact of DA frequency on non-Gaussianity is investigated by means of the relative KLD difference between the 5MIN and all the other experiments, computed as:

$$\overline{KLD_{diff}} = \frac{\overline{KLD_E} - \overline{KLD_{5MIN}}}{\overline{KLD_{5MIN}}}$$

where $\overline{KLD_{diff}}$ is the relative difference between the averaged KLD in the 5MIN experiment ($\overline{KLD_{5MIN}}$) and on each of the other experiments ($\overline{KLD_E}$), where E can be either 5MIN-4D, 2MIN, 1MIN, 1MIN-4D or 30SEC).

5. Please explain "closest" quantitatively.

Following the reviewer's comment we expand the discussion to better explain possible time differences between the observations and the analysis time.
Here, only a single volume scan closest to the analysis time is used per analysis. Namely, more frequent updates assimilate more data. In all cases the time difference between observation time (center time of the radar volume scan) and the analysis time do not differ by more than 15 seconds.

6. The higher the update frequency, will it break the balance between physical variables? How to understand and explain.

We agree with the reviewer that this is an important aspect that has been overlooked in the previous version of the manuscript. In the revised version we include a discussion about the effect of imbalance.

As has been shown in the previous studies, more frequent assimilation can produce a larger degree of imbalance in the initial conditions which can degrade the quality of the forecasts (e.g. Lange and Craig 2014, Bick et al. 2016). Therefore, despite the potential benefits of a more Gaussian model error distribution on the analysis accuracy, other factors may degrade the forecasts initialized from more frequent data assimilation cycles. Imbalance and non-Gaussianity can also possibly be related. Gaussian error distributions can lead to more physically meaningful assimilation updates in the context of an EnKF and thus more balanced initial conditions. However, a larger imbalance in the initial conditions can contribute to faster error growth and increased departure from the Gaussian in the forecast distribution. Possible interactions of these mechanisms in a data assimilation cycle have not been investigated, and are a subject for future research. Our results suggest that despite the effect of a larger imbalance, the increase of DA frequency reduces non-Gaussianity in the sample distributions obtained with the EnKF. This is even true for variables like the vertical velocity within convective clouds which are frequently used to measure the effect of imbalance in the initial conditions.

Added references:
Lange, H. and Craig, G. C.: The impact of data assimilation length scales on analysis and prediction of convective storms, Mon. Wea. Rev.,
142, 3781–3808, 2014.

Bick, T., Simmer, C., Trömel, S., Wapler, K., Stephan, K., Blahak, U., Zeng, Y., and Potthast, R.: Assimilation of 3D-Radar Reflectivities with an Ensemble Kalman Filter on the Convective Scale, Quart. J. Roy. Meteor. Soc., 142, 1490–1504, 2016.

7. This paper used the super-observations. What is the relationship between observation scale and grid scale matching and more observations?

We agree with the reviewer in that this point deserves further clarification. We expand the discussion in the paper. Reflectivity and Doppler velocity observations are superobbed to horizontal resolution of 1 km and vertical resolution of 500 m to approximately match the model resolution. This is done to reduce the amplitude of spatial scales, in the observed data, which are not well resolved by the model as well as that of small scale noise. This procedure can also reduce the impact of possible spatial correlations in the observation errors.

8. This article needs polishing.

Based on the reviewer's comment and on the comment of the other two reviewers we perform an extended revision and polishing of the text and improvement of the figures. We hope that these changes improved the overall presentation quality of the manuscript.

---

## Author Comment (AC2)

**Reviewer #2**

Review of "Reduced non-Gaussianity by 30-second rapid update in convective-scale numerical weather prediction " by J. Ruiz et al..

General comments:

This paper has investigated how the DA frequencies affect non-Gaussianity using a high resolution NWP model with LETKF method for data assimilation. DA experiments with different frequencies are conducted using real observations. They have some findings about the non-Gaussianity in data assimilation, which are quite new and interesting. They have used a high resolution DA system with very high DA frequencies to support their conclusions. And they have analyzed the results comprehensively. The manuscript is overall well-written. I am in support of publishing this manuscript after minor revision.

We would like to thank the reviewer for all the suggestions on how to improve and expand the discussion. Based on the reviewer's comments we add important additional explanations and discussions.

Specific comments:

1. This work measures the non-Gaussianity (by KLD) of the analysis fields. They found that increasing the assimilation frequency up to 30 seconds and assimilating more observations can reduce KLD. This conclusion can be expected easily. As acknowledged, EnKF and LETKF are sub-optimal when the forecast error are non-Gaussian, and the non-Gaussianity of the forecast error grows during model integration. If the DA frequency is higher, the non-Gaussianity of the forecast will be smaller due to shorter integration period, therefore the EnKF will be more effective.

However, they didn't show the KLD of the prior error distribution. If they can compare the posterior KLD with prior KLD with different DA frequencies, they can better illustrate the "reduced non-Gaussianity by 30-second rapid update" in the title.

We agree with the reviewer and add a new figure (Fig. 7) to better illustrate this point. This figure provides a better insight on the effect of data assimilation on the non-Gaussianity of the error distribution and shows some interesting differences between the impact of the assimilation in the "raining" and "non-raining" grid points. We also include a discussion in the revised version of the paper.

To investigate the effect of the analysis update on non-Gaussianity we present the time series of the KLD of the analysis and first guess vertically and horizontally averaged over the "raining" and "non-raining" grid points (Fig. 7). At most times and variables over the "raining" and "non-raining" grid points KLD is reduced during the assimilation step. Experiments with longer windows experience more KLD growth during the forecast as expected, but also a larger reduction at the analysis step, which is not as effective as the more frequent updates in reducing the analysis KLD. As noted before, the specific humidity over the "non-raining"

grid points behaves differently, in this case, KLD increases during the assimilation step for almost all times and experiments leading to larger KLD at shorter assimilation windows (Figs. 6b,f of the submitted manuscript). In this area mostly "non-raining" observations are assimilated to suppress spurious clouds. Interestingly in the "non-raining" grid points 5MIN-4D is the experiment providing the lowest KLD for all variables (Figs. 7b,d,f). This result suggests the potential benefits of treating "non-raining" observations differently.

---

## Author Comment (AC3)

**Reviewer #3**

General comments:

This study focuses on investigating the impacts of DA update frequency and observation number on the non-Gaussianity of model simulation error, in a case of strong convection. Results shown in the manuscript show that the non-Gaussianity of error can be reduced by increasing the DA frequency and number of observations, which could possibly improve the performance of EnKF. While the results are impressive, there are several problems the authors may need to address before the manuscript is published. I hereby recommend a major revision.

We would like to thank Reviewer #3 for raising interesting points that helped expand the discussion and to add interesting results. Also the comments helped to clarify several aspects that were not clear in the original version of the manuscript.

Specific comments:

1. Model configuration: The authors did not provide enough information about the model's configurations. In line 59, the model used in this study has a horizontal resolution of 1 km, 50 vertical sigma levels, and a size of 180 km by 180 km (Fig. 1a). I wonder what the range and resolution of the vertical sigma levels are defined. According to my knowledge, models with higher horizontal resolutions should also have higher vertical resolutions. The number of vertical levels of the model introduced in this study is probably too coarse for 1km-scale simulations. (also see specific comment 2)

There were missing important configuration aspects in the original version of the manuscript. To address this point we add more details about the vertical discretization. 50 vertical levels extend up to 18 km elevation with a variable grid spacing from 140 m to 790 m in a hybrid sigma-z terrain-following coordinate.

2. According to the paper, better results of radar data assimilation were obtained with vertical localization scale of 2 km and horizontal localization of 4 km (Line 70 – 73). In this sense, a 1:2 ratio of the horizontal to vertical resolutions of the model could give more reliable simulation results. i.e., If, in this study, the model's horizontal resolution if set as 1 km, then the vertical resolution could be set as around 500 m.

We agree with the reviewer. The vertical resolution is variable with higher resolution close to the surface. On average the vertical resolution is approximately 360 m which satisfies this condition. The vertical grid spacing is less than 500 m up to 8-km height, then it becomes more than 500 m beyond 8-km height and reaches 790 m at the model top.

3. The authors mentioned that the non-Gaussianity reduced by 40% when assimilation window length shortened from 5 minutes to 30 seconds. What are the main benefits from the reduction? The authors claimed that this could improve the performance of the EnKF, without showing any evidence. It might be better by simply showing the error of precipitation output simulated in different experiments.

We would like to thank the reviewer for raising this important point. Following this comment we include a new figure (Figure 8) showing the RMSE and bias of the analysis and first guess with respect to the observed maximum reflectivity. We also include a discussion of the results.

To evaluate the impact of assimilation frequency on the distance between the analysis and first guess to the observations in a more systematic way, we compute the root mean squared error (RMSE) and bias for reflectivity observations (Fig. 8). The computation of the RMSE and bias between the model and the observations is done by comparing the column maximum of the reflectivity for each horizontal grid location and time. The RMSE and bias are computed only over grid points at which the observed maximum reflectivity is over 5 dBZ. The time series of RMSE shows a better fit to the observed reflectivity for shorter assimilation windows. The impact of 4D DA is not so clear, 1MIN-4D slightly outperforms the 1MIN but 5MIN-4D and 5MIN perform similarly (Fig. 7a). This is partially because in 4D data assimilation the analysis results from the assimilation of all the observations within the assimilation window, while to construct this figure, only the observations at the analysis time were considered. The bias, computed as the mean difference between the model and the observations does not seem to be consistently affected by the assimilation frequency (Fig. 8b). These results are in agreement with those observed in the time series of KLD for different variables. However, we should be cautious with the interpretation of these results since increasing the observation number can lead to both a reduced KLD and a better fit to the observed quantities, not necessarily implying a causal link between these two effects.

[Figure]

Figure 8: Sawtooth time-series of the root mean squared error (dBZ, a) and bias (dBZ, b) of the maximum reflectivity of the analysis and first guess for the 5MIN (red), 5MIN-4D (blue), 2MIN (green), 1MIN (magenta), 1MIN-4D (black) and 30SEC (cyan) experiments.

4. 2e–h: The authors marked the location of the maximum KLD for vertical velocity at the lower troposphere in Fig. 2f, but middle troposphere in Figs. 2e, 2g and 2h even though a maximum KLD center is not obvious in Figs. 2g and 2h. If the authors intended to emphasize the improvement of KLD in the middle troposphere, they should consider the KLD in the middle troposphere in all cases.

We would like to thank the reviewer for pointing this out. The explanation in the previous version of the manuscript was not clear. Our intention is not to focus on the middle troposphere but to show examples of distributions associated with KLD maxima within convective clouds.

We restrict the search of the maximum KLD to the grid points at which the ensemble mean reflectivity is over 30 dBZ to investigate departures from the Gaussian within convective clouds where radar data impact is larger.

5. 5b and 5h: The main highlight of this figure (which is also that of the manuscript) is improvement in the KLD with higher DA frequency. However, Fig. 5 also shows obvious increase in the KLD of specific humidity in the 30SEC experiment, in both the raining and non-raining cases. And, it seems that the authors did not make any discussion on these results. While there are great improvements in KLD of most grid points, especially that of vertical velocity when the authors focus on convective-scale simulation, why is the same improvement not obtained for specific humidity? Are errors generated from the more frequent DA update?

We thank the reviewer for bringing this interesting point. We add a remark to the discussion in Figure 6 (previously Figure 5), about the different impacts of frequent updates on specific humidity and other variables.

There are some exceptions to the general reduction in non-Gaussianity with increased update frequency. Specific humidity in non-raining grid points shows larger KLD in the 30SEC than in the 5MIN experiments. This is also the case for the raining grid points at upper levels in the second half of the experiment. Also the KLD in W in the non-raining grid points at middle and upper levels is slightly larger in the 30SEC experiment.

We also analyse this in more detail in Figure 7 which is a new figure showing the impact of the forecast and the assimilation step on the KLD.

To investigate the effect of the analysis update on non-Gaussianity we present the time series of the KLD of the analysis and first guess vertically and horizontally averaged over the "raining" and "non-raining" grid points (Fig. 7). At most times and variables over the "raining" and "non-raining" grid points KLD is reduced during the assimilation step. Experiments with longer windows experience more KLD growth during the forecast as expected, but also a larger reduction at the analysis step, which is not as effective as the more frequent updates in reducing the analysis KLD. As noted before, the specific humidity over the "non-raining" grid points behaves differently, in this case, KLD increases during the assimilation step for almost all times and experiments leading to larger KLD at shorter assimilation windows (Figs. 7b,f). In this area mostly "non-raining" observations are assimilated to suppress spurious clouds. Interestingly in the "non-raining" grid points 5MIN-4D is the experiment providing the lowest KLD for all variables (Figs. 7b, d and f).

[Figure]

Figure 7: Sawtooth time-series of the KLD ($10^{-2}$) of the analysis and first guess over the rainy (<0dBZ, a,c,e) and non-rainy (>30dBZ, b,d,f) grid points for temperature (a,b), specific humidity (c,d) and vertical velocity (e,f) and for the 5MIN (red), 5MIN-4D (blue), 2MIN (green), 1MIN (magenta), 1MIN-4D (black) and 30SEC (cyan) experiments.

6. Line 67: "… Climate Forecast System Reanalysis Saha et al. (2010)" à "… Climate Forecast System Reanalysis (Saha et al., 2010)"

We change this following the reviewer's comment.

7. Line 99 and Eq.(1): "where P(x) and Q(x) are two . . . " à should be "where p(x) and q(x) are two . . ."?

We agree with the reviewer and change the sentence accordingly.

8. Lines 140–141: Wrong use of "so that".

Following the reviewer's suggestion, we change the sentences in the following way: "However, KLD in 5MIN-4D is larger than that in 30-SEC or 1MIN-4D, **indicating that** DA frequency is equally important."

9. Lines 144: "raining" and "non-raining" grid points sound better than "rain" and "non-rain" grid points to me.

We change the definition to "raining" and "non-raining" as suggested by the reviewer.

---

## Author Response (AR1)

**Reviewer #1**

Review of "Reduced non-Gaussianity by 30-second rapid update in convective-scale numerical weather prediction" by J. Ruiz et al..

General comments: The manuscript investigated the degree of non-Gaussianity of forecast error distributions and how it is affected by the DA update frequency and observation number. This article used 1000 ensembles, but the generation of ensembles is not very clear. The introduction of the KLD method is not clear. Some words are not rigorous. This paper theoretically provides evidence that increasing the update frequency and the observations can improve the accuracy of assimilation in convective-scale. I think this manuscript can be considered for publication if these concerns could be addressed:

We would like to thank the reviewer for the comments which help to improve the clarity of the presentation and also bring additional discussions to the paper. Based on the reviewer's comments we expand the description of the methodology and add additional explanations and discussions.

Specific comments:

1. The generation of ensembles need to be described in detail, because it is very important for this article and the ensemble DA method.

"The initial ensemble at the first assimilation cycle and the boundary condition ensemble are created by adding random perturbations which preserve the hydrostatic and nearly geostrophic equilibrium (Necker et al. 2020, Maldonado et al. 2021). These perturbations are generated from a sample of continuous 6-hourly analysis states provided by the Climate Forecast System Reanalysis (CFSR, Saha et al. 2020, $[X_{CFSR}(t_1), X_{CFSR}(t_1),...,X_{CFSR}(t_N)]$, where N=5840 (4 years). The horizontal grid spacing of the CFSR data is 0.5°. At the beginning of the assimilation cycle ($t=t_s$), the initial condition perturbation of the i-th member $X'^{(i)}$ is computed as:

$$X'^{(i)}(t_s) = \alpha \left[ X_{CFSR}(t_{n_1{}^{(i)}}) - X_{CFSR}(t_{n_2{}^{(i)}}) \right]$$

where $\alpha$ is a multiplicative factor equal to 0.1 so that the amplitude of the perturbations is roughly equivalent to 10% of the climatological variability. The two CFSR analysis states are chosen by randomly selecting two numbers $n_1{}^{(i)}$ and $n_2{}^{(i)}$ from the N elements satisfying the condition that $t_{n1(i)}$ and $t_{n2(i)}$ correspond to the same time of the year and time of the day. In the following assimilation cycles at time $t>t_s$, we obtain the boundary perturbations as:

$$X'^{(i)}(t) = \alpha\left[(1-\beta)\Big(X(t_{l_1{}^{(i)}}) - X(t_{l_2{}^{(i)}})\Big) + \beta\Big(X(t_{u_1{}^{(i)}}) - X(t_{u_1{}^{(i)}})\Big)\right]$$

where $l^{(i)}_{1,2} = n_{1,2}^{(i)} + m$ and $u^{(i)}_{1,2} = n_{1,2}^{(i)} + m + 1$, with $m$ = floor[$(t - t_s)/6$ h] and $\beta$ = [$(t-t_s)/6$ h] - $m$ being a temporal linear interpolation factor to compute perturbations at arbitrary times (not necessarily a multiple of 6h). In this way we obtain perturbations that are smoothly varying in time and consistently with the large scale dynamics of the atmosphere. This procedure is applied to all atmospheric and soil state variables."

2. The introduction of KLD should contain how to operate in this article? What should be noted? So that readers can repeat your experiment

We agree with the reviewer and we expanded the description on how KLD is computed in this work. In particular, we added more details on how KLD is computed from the ensemble-based sample, and an equation expressing how KLD is computed in this paper. These descriptions have been added starting on Line 111.

"To measure the degree of non-Gaussianity of the error distributions we compute the Kullback-Leibler divergence (hereafter KLD, Kullback 1951) which is defined as follows:

$$KLD(P\|Q) = \int_{-\infty}^{\infty} p(x)\ln\frac{p(x)}{q(x)}dx,$$

where $p(x)$ and $q(x)$ are the probability density functions (PDFs) of P and Q respectively. The KLD is 0 if P and Q are the same and takes positive values if P and Q differ. In our case $p(x)$ is either the first guess or analysis error distribution for the state variable x, and $q(x)$ is a Gaussian distribution whose mean and standard deviation are equal to the ones of $p(x)$. Therefore, a low KLD value corresponds to the first guess or analysis error distribution close to a Gaussian.
In the EnKF we do not have access to the continuous PDF $p(x)$ but to its finite, limited sample. For each state variable x (e.g. temperature, wind components, etc) and at each model grid point, we approximate $p(x)$ with the sample histogram from the 1000-member ensemble using 32 equally-sized bins covering the range where $p(x)$ is greater than 0. This range is defined by the minimum and maximum values of x at each model grid point and time. Hence, we can approximate the KLD as follows:

$$KLD(P\|Q) \approx \sum_{j=1}^{j=32} p_j\ln\frac{p_j}{q_j},$$

where $p_j$ is the empirical frequency of x at the j-th histogram bin. $q_j$ is the integral over the j-th histogram bin of a Gaussian PDF whose mean and standard deviation are given by the ensemble-based sample estimates. After implementing this we end up with an estimation of the KLD of the analysis and first guess error distributions with respect to the Gaussian for each grid point location, vertical level, and time."

3. Is the solution of KLD grid point by grid ? Using the assimilated ensembles to statistics?

Following this comment and the previous one we extend the description on how KLD is computed so that the experiments can be reproduced. The specifications on how KLD is computed from the ensemble members is included starting on Line 111 (see also the answer corresponding to the previous comment).

4. Please give the formula of relative KLD difference.

We include an equation for the relative KLD difference. This equation and its description has been included in Line 177 of the revised version of the manuscript.

"The impact of DA frequency on non-Gaussianity is investigated by means of the relative KLD difference between the 5MIN and all the other experiments, computed as:

$$\overline{KLD_{diff}} = \frac{\overline{KLD_E} - \overline{KLD_{5MIN}}}{\overline{KLD_{5MIN}}}$$

where $\overline{KLD_{diff}}$ is the relative difference between the averaged KLD in the 5MIN experiment ($\overline{KLD_{5MIN}}$) and on each of the other experiments ($\overline{KLD_E}$), where E can be either 5MIN-4D, 2MIN, 1MIN, 1MIN-4D or 30SEC)."

5. Please explain "closest" quantitatively.

Following the reviewer's comment we expand the discussion to better explain possible time differences between the observations and the analysis time. The following comment has been expanded, starting on Line 103:
"Here, only a single volume scan closest to the analysis time is used per analysis. Namely, more frequent updates assimilate more data. In all cases the time difference between the observation time (center time of the radar volume scan) and the analysis time do not differ by more than 15 seconds. "

6. The higher the update frequency, will it break the balance between physical variables? How to understand and explain.

We agree with the reviewer that this is an important aspect that has been overlooked in the previous version of the manuscript. In the revised version we include a discussion about the effect of imbalance in the "Summary and discussion" section (starting on Line 267):

"Moreover, as has been shown in the previous studies, more frequent assimilation can produce a larger degree of imbalance in the initial conditions which can degrade the quality of the forecasts (e.g. Lange and Craig 2014, Bick et al. 2016). Therefore, despite the potential benefits of a more Gaussian model error distribution on the analysis accuracy, other factors may degrade the forecasts initialized from more frequent data assimilation cycles. Imbalance may also be an additional source of non-Gaussianity. Gaussian error distributions can lead to more physically meaningful assimilation updates in the context of an EnKF and therefore, more balanced initial conditions. However, a larger imbalance in the initial conditions can contribute to faster error growth and increased departure from the Gaussian in the forecast distribution. Possible interactions of these mechanisms in a data assimilation cycle have not been investigated, and are a subject for future research. Our results suggest that despite the effect of a larger imbalance, the increase of DA frequency reduce non-Gaussianity in the sample distributions with the EnKF. This is even true to variables like vertical velocity within convective clouds which are frequently used to measure the effect of imbalance in the initial conditions. "

Added references:
Lange, H. and Craig, G. C.: The impact of data assimilation length scales on analysis and prediction of convective storms, Mon. Wea. Rev.,
142, 3781–3808, 2014.

Bick, T., Simmer, C., Trömel, S., Wapler, K., Stephan, K., Blahak, U., Zeng, Y., and Potthast, R.: Assimilation of 3D-Radar Reflectivities with an Ensemble Kalman Filter on the Convective Scale, Quart. J. Roy. Meteor. Soc., 142, 1490–1504, 2016.

7. This paper used the super-observations. What is the relationship between observation scale and grid scale matching and more observations?

We agree with the reviewer in that this point deserves further clarification. We expand the discussion as follows (See Line 91 in the revised version of the manuscript):

"Reflectivity and Doppler velocity observations are superobbed to horizontal resolution of 1 km and vertical resolution of 500 m to approximately match the model resolution. This helps reduce the errors of representativeness due to the gap between what is represented by the model and observation. This procedure can also reduce the impact of possible spatial correlations in the observation errors."

8. This article needs polishing.

Based on the reviewer's comment and on the comment of the other two reviewers we perform an extended revision and polishing of the text and improvement of the figures. We hope that these changes improved the overall presentation quality of the manuscript.
**Reviewer #2**

Review of "Reduced non-Gaussianity by 30-second rapid update in convective-scale numerical weather prediction " by J. Ruiz et al..

General comments:

This paper has investigated how the DA frequencies affect non-Gaussianity using a high resolution NWP model with LETKF method for data assimilation. DA experiments with different frequencies are conducted using real observations. They have some findings about the non-Gaussianity in data assimilation, which are quite new and interesting. They have used a high resolution DA system with very high DA frequencies to support their conclusions. And they have analyzed the results comprehensively. The manuscript is overall well-written. I am in support of publishing this manuscript after minor revision.

We would like to thank the reviewer for all the suggestions on how to improve and expand the discussion. Based on the reviewer's comments we add important additional explanations and discussions.

Specific comments:

1.  This work measures the non-Gaussianity (by KLD) of the analysis fields. They found that increasing the assimilation frequency up to 30 seconds and assimilating more observations can reduce KLD. This conclusion can be expected easily. As acknowledged, EnKF and LETKF are sub-optimal when the forecast error are non-Gaussian, and the non-Gaussianity of the forecast error grows during model integration. If the DA frequency is higher, the non-Gaussianity of the forecast will be smaller due to shorter integration period, therefore the EnKF will be more effective.
    However, they didn't show the KLD of the prior error distribution. If they can compare the posterior KLD with prior KLD with different DA frequencies, they can better illustrate the "reduced non-Gaussianity by 30-second rapid update" in the title.

We agree with the reviewer and add a new figure (Fig. 7 in the revised version of the manuscript) to better illustrate this point. This figure provides a better insight on the effect of data assimilation on the non-Gaussianity of the error distribution and shows some interesting differences between the impact of the assimilation in the "precipitating" and "non-precipitating" grid points. We also include this discussion in the text (starting on Line 205):

"To investigate the effect of the analysis update on non-Gaussianity we present the time series of the KLD of the analysis and first guess vertically and horizontally averaged over the "precipitating" and "non-precipitating" grid points (Fig. 7). At most times and variables over the "precipitating" and "non-precipitating" grid points, KLD is reduced during the assimilation step. Experiments with longer windows show more KLD growth during the forecast as expected, but also a larger reduction at the analysis step, which is not as effective as the more frequent updates in reducing the analysis KLD. As noted before, the specific humidity over the "non-precipitating" grid points behaves differently, and KLD increases during the assimilation step for almost all times and experiments, leading to larger KLD at shorter assimilation windows (Figs. 6b, f). In this area mostly "non-precipitating" observations are assimilated to suppress spurious clouds. Interestingly in the "non-precipitating" grid points 5MIN-4D is the experiment providing the lowest KLD for all variables (Figs. 7b, d and f). This result suggests the potential benefits of treating "non-precipitating" observations differently."

[Figure]

Figure 7: Sawtooth time-series of the KLD $(10^{-2})$ of the analysis and first guess horizontally and vertically averaged over the precipitating (<0dBZ, a,c,e) and non-precipitating (>30dBZ,

b,d,f) grid points for temperature (a,b), specific humidity (c,d) and vertical velocity (e,f) and for the 5MIN (red), 5MIN-4D (blue), 2MIN (green), 1MIN (magenta), 1MIN-4D (black) and 30SEC (cyan) experiments.

2. Page 5, lines 110-125 and figure 2. Compared with the rest of this article, the readability of this paragraph is poor. They have shown too much information in figure 2, such that they need to use parentheses constantly to indicate the subplots and features (shades or contours) in figure 2. And there are also some problems with the order of expression in this paragraph, therefore the readers have to look at the subplots back and forth. I suggest splitting the paragraph from line 116 or 117.

We agree with this comment. To improve the readability of this section we split the original Figure 2 into 2 figures (Figs. 2 and 3 of the new version of the manuscript). We also reorder the discussion to reduce the need to go back and forward from one figure to the next. We hope that these changes helped to improve the clarity of the discussion.

3. Figure 2e-h, they use shades to show KLD for W, while use blue contours for KLD for T and red contours for ensemble spread. This is very odd. In my opinion, use contours of different colors to show KLD for different variables seems more reasonable.

We agree with the reviewer's comment, Figure 2 in the original version of the manuscript, contains too many lines of different colors in the same panel. To address this, we add a new row in Figures 2 and 3 (Figures 2 and 4 in the revised version of the manuscript) to separate the KLD and ensemble spread for W and T. Now the KLD is shown in shaded for both variables and the ensemble spread is shown in contours. Following is the new version of Figure 2:

[Figure]

Figure 2: (a-h) South-North vertical cross-section along the black line indicated in Fig. 1b-d at 0530 UTC for (a-d) first-guess ensemble-mean reflectivity (Z, shades, dBZ) and vertical velocity (W, contours every 2.5 ms$^{-1}$), (e-h) vertical velocity KLD (shades, 10$^{-2}$) and ensemble spread (red contours at 1.0, 2.5, 5.0 and 10.0 ms$^{-1}$), and (i-l) temperature KLD (shades, 10$^{-2}$) and ensemble spread (red contours at 0.2, 0.5, 1.0 and 2.0 K). Blacked dashed contours indicate reflectivity over 30 dBZ. The black cross in panels (i-l) indicates the location of the maximum KLD within the grid points at which Z > 30dBZ.

4. Figure 2a-h, the location of the maximum KLD for vertical velocity is shown by blue circles. I think the circle is too large and its color is inappropriate. I cannot clearly see whether the ensemble spread maxima are slightly out of phase with respect to the KLD maxima. What about a black x or plus sign?

We agree with this comment. Following the reviewer's suggestion we replace the blue circle by a black cross (See new version of Figure 2 in the answer to the previous comment). Also by reducing the number of contours in the same panel, now the cross is more visible. We also added more detail on to what extent the distribution of the

ensemble spread can be associated to that of the KLD in this particular case. The discussion has been modified starting on Line 144:

"Kondo and Miyoshi (2019) found that in synoptic scales, the ensemble spread maxima are collocated with the KLD maxima. At convective scales for W, the ensemble spread maxima (Fig. 2e, red contours) are not necessarily collocated. For example, larger departures from the Gaussian are found above the ensemble spread maximum associated with the main updraft in the 5MIN experiment. For temperature also there is no clear relation in the distribution of the ensemble spread and the KLD, although KLD maxima seem to occur within areas of relatively large ensemble spread. As the assimilation frequency increases it is more difficult to find a relationship between KLD and ensemble spread either for W or T (Fig. 2 second and third rows)."

5. Line 114 and line 117. The authors have shown that "KLD is reduced more from 5MIN to 2MIN than from 1MIN to 30SEC" and "The ensemble spread for W is reduced significantly from 5MIN to 2MIN". I think this is also associated with the nonlinearity of this model. Could it possible that the non-Gaussianity of prior distribution grows fastest during the freerun between 2min to 5min?

We would like to thank the reviewer for raising this interesting point. We agree with this hypothesis and we add a discussion starting Line 247 in the revised version of the manuscript. To the best of our knowledge there are no studies dealing with error growth at these short time scales which could be an interesting investigation that can address this issue in detail.

"There are two possible ways in which more frequent DA can result in error distributions closer to the Gaussian. First, more frequent DA contributes to a quasi-linear evolution of the forecast error due to forecast lengths which are shorter than the predictability limit for the resolved scales. This also helps keep the perturbation small and can additionally contribute to quasi-linear perturbation dynamics. Second, our results show that the analysis step effectively contributes to reducing non-Gaussianity for different variables, although this may not be the case for "non-precipitating" reflectivity observations that produce an increase in KLD for specific humidity. Non-gaussianity reduction during DA is larger with longer windows. However, it is not enough to compensate for the effect of more rapid and non-linear error growth during the forecast step in the lower update frequency experiments.

From the point of view of KLD reduction, the largest impact is found between 5MIN and 2MIN updates. This suggests that non-linear error growth become more important after the first 2 minutes of integration at these scales. This hypothesis is partially supported by the reduction in RMSE and ensemble spread. A 2-minute update frequency seems to provide a good compromise between the computational cost and non-Gaussianity of the error distributions. However, from the point of view of the analysis accuracy more frequent DA provides a better fit to the observed quantities. The specific role of reduced non-Gaussianity on this is not clear and should be further investigated. Gaussian error distributions may contribute to more accurate analysis updates, but in the current experimental setting, other

factors like the increase in the number of assimilated observations may also lead to the reduction in the RMSE for observed quantities. Maejmima and Miyoshi 2020 investigated the impact of assimilation frequency at 1-km using observing system simulation experiments. They also found a significant improvement in the forecast quality when the assimilation window is reduced from 5 minutes to 3 minutes and additional improvements using 1 minute windows. These results are consistent with what is found in this paper with respect to Gaussianity in the error distributions."

6. Since increasing the analysis update frequency from 5 minutes to 2 minutes has most significant impact upon non-Gaussianity. So can we say it is a optimal strategy considering the trade-off between cost and efficiency?

We agree that this would be a good compromise in terms of computational cost and non-Gaussianity. However, other aspects have to be taken into account like for example the impact of update frequency on the imbalance in the initial conditions (also mentioned by reviewer #1) which can negatively affect the quality of the forecast. In this work we investigate the impact on non-Gaussianity which can contribute to improving the quality of the initial conditions with more frequent updates, but in the context of a data assimilation cycle other aspects have to be taken into account. We include part of this discussion in the answer to the previous comment, but we also add a caution following this discussion. We include a discussion starting on Line 267 of the revised version of the manuscript.

"Moreover, as has been shown in the previous studies, more frequent assimilation can produce a larger degree of imbalance in the initial conditions which can degrade the quality of the forecasts (e.g. Lange and Craig 2014, Bick et al. 2016). Therefore, despite the potential benefits of a more Gaussian model error distribution on the analysis accuracy, other factors may degrade the forecasts initialized from more frequent data assimilation cycles. Imbalance may also be an additional source of non-Gaussianity. Gaussian error distributions can lead to more physically meaningful assimilation updates in the context of an EnKF and therefore, more balanced initial conditions. However, a larger imbalance in the initial conditions can contribute to faster error growth and increased departure from the Gaussian in the forecast distribution. Possible interactions of these mechanisms in a data assimilation cycle have not been investigated, and are a subject for future research. Our results suggest that despite the effect of a larger imbalance, the increase of DA frequency reduce non-Gaussianity in the sample distributions with the EnKF. This is even true to variables like vertical velocity within convective clouds which are frequently used to measure the effect of imbalance in the initial conditions. "

Added references:
Lange, H. and Craig, G. C.: The impact of data assimilation length scales on analysis and prediction of convective storms, Mon. Wea. Rev.,
142, 3781−3808, 2014.

Bick, T., Simmer, C., Trömel, S., Wapler, K., Stephan, K., Blahak, U., Zeng, Y., and Potthast, R.: Assimilation of 3D-Radar Reflectivities
with an Ensemble Kalman Filter on the Convective Scale, Quart. J. Roy. Meteor. Soc., 142, 1490–1504, 2016."

**Reviewer #3**

General comments:

This study focuses on investigating the impacts of DA update frequency and observation number on the non-Gaussianity of model simulation error, in a case of strong convection. Results shown in the manuscript show that the non-Gaussianity of error can be reduced by increasing the DA frequency and number of observations, which could possibly improve the performance of EnKF. While the results are impressive, there are several problems the authors may need to address before the manuscript is published. I hereby recommend a major revision.

We would like to thank Reviewer #3 for raising interesting points that helped expand the discussion and to add interesting results. Also the comments helped to clarify several aspects that were not clear in the original version of the manuscript.

Specific comments:

1. Model configuration: The authors did not provide enough information about the model's configurations. In line 59, the model used in this study has a horizontal resolution of 1 km, 50 vertical sigma levels, and a size of 180 km by 180 km (Fig. 1a). I wonder what the range and resolution of the vertical sigma levels are defined. According to my knowledge, models with higher horizontal resolutions should also have higher vertical resolutions. The number of vertical levels of the model introduced in this study is probably too coarse for 1km-scale simulations. (also see specific comment 2)

There were missing important configuration aspects in the original version of the manuscript. To address this point we add the following sentence in Line 59 of the revised version of the manuscript:

"50 vertical levels extend up to 18 km elevation with a variable grid spacing from 140 m to 790 m in a hybrid sigma-z terrain-following coordinate."

2. According to the paper, better results of radar data assimilation were obtained with vertical localization scale of 2 km and horizontal localization of 4 km (Line 70 – 73). In this sense, a 1:2 ratio of the horizontal to vertical resolutions of the model could give more reliable simulation results. i.e., If, in this study, the model's horizontal resolution if set as 1 km, then the vertical resolution could be set as around 500 m.

We agree with the reviewer. The vertical resolution is variable with higher resolution close to the surface. On average the vertical resolution is approximately 360 m which satisfies this condition. The vertical grid spacing is less than 500 m up to 8-km height, then it becomes more than 500 m beyond 8-km height and reaches 790 m at the model top.

3. The authors mentioned that the non-Gaussianity reduced by 40% when assimilation window length shortened from 5 minutes to 30 seconds. What are the main benefits from the reduction? The authors claimed that this could improve the performance of the EnKF, without showing any evidence. It might be better by simply showing the error of precipitation output simulated in different experiments.

We would like to thank the reviewer for raising this important point. Following this comment we include a new figure (Figure 8 in the revised version of the manuscript) showing the RMSE and bias of the analysis and first guess with respect to the observed maximum reflectivity. We also include a discussion of the results starting on Line 215.

"To evaluate the impact of assimilation frequency on the distance between the analysis and first guess to the observations in a more systematic way, we compute the root mean squared error (RMSE) and bias for reflectivity observations (Fig. 8). The computation of the RMSE and bias between the model and the observations is done by comparing the column maximum of the reflectivity for each horizontal grid location and time. The RMSE and bias are computed only over grid points at which the observed maximum reflectivity is over $5$ $dBZ$. The time series of RMSE shows a better fit to the observed reflectivity for shorter assimilation windows. The impact of 4D DA is not so clear, 1MIN-4D slightly outperforms the 1MIN but 5MIN-4D and 5MIN perform similarly (Fig. 8, a). This is partially because in 4D data assimilation the analysis results from the assimilation of all the observations within the assimilation window, while to construct this figure, only the observations at the analysis time were considered. The bias, computed as the mean difference between the model and the observations does not seem to be consistently affected by the assimilation frequency (Fig. 8, b). These results are in agreement with those observed in the time series of KLD for different variables. However, we should be cautious with the interpretation of these results since increasing the observation number can lead to both a reduced KLD and a better fit to the observed quantities, not necessarily implying a causal link between these two effects."

[Figure]

Figure 8:Sawtooth time-series of the root mean squared error (dBZ, a) and bias (dBZ, b) of the maximum reflectivity of the analysis and first guess for the 5MIN (red), 5MIN-4D (blue), 2MIN (green), 1MIN (magenta), 1MIN-4D (black) and 30SEC (cyan) experiments.

4.  2e–h: The authors marked the location of the maximum KLD for vertical velocity at the lower troposphere in Fig. 2f, but middle troposphere in Figs. 2e, 2g and 2h even though a maximum KLD center is not obvious in Figs. 2g and 2h. If the authors intended to emphasize the improvement of KLD in the middle troposphere, they should consider the KLD in the middle troposphere in all cases.

We would like to thank the reviewer for pointing this out. The explanation in the previous version of the manuscript was not clear. Our intention is not to focus on the middle troposphere but to show examples of distributions associated with KLD maxima within convective clouds. We add the following sentence in Line 157 to clarify this point:

"We restrict the search of the maximum KLD to the grid points at which the ensemble mean reflectivity is over 30 dBZ where radar data impact would be large."

5.  5b and 5h: The main highlight of this figure (which is also that of the manuscript) is improvement in the KLD with higher DA frequency. However, Fig. 5 also shows obvious increase in the KLD of specific humidity in the 30SEC experiment, in both the raining and non-raining cases. And, it seems that the authors did not make any discussion on these results. While there are great improvements in KLD of most grid points, especially that of vertical velocity when the authors focus on convective-scale simulation, why is the same improvement not obtained for specific humidity? Are errors generated from the more frequent DA update?

We thank the reviewer for bringing this interesting point. We first add this remark to the discussion in Figure 6 (previously Figure 5), about the different impacts of frequent updates on specific humidity and other variables (starting on Line 201):

"There are some exceptions to the general reduction in non-Gaussianity with increased update frequency. Specific humidity in non-precipitating grid points shows larger KLD in the 30SEC than in the 5MIN experiments. This is also the case for the precipitating grid points at upper levels in the second half of the experiment. Also the KLD in W in the non-precipitating grid points at middle and upper levels is slightly larger in the 30SEC experiment."

We also analyse this in more detail in Figure 7 which is a new figure showing the impact of the forecast and the assimilation step on the KLD with the following discussion starting on Line 205 of the revised version of the manuscript:

"To investigate the effect of the analysis update on non-Gaussianity we present the time series of the KLD of the analysis and first guess vertically and horizontally averaged over the "precipitating" and "non-precipitating" grid points (Fig. 7). At most times and variables over the "precipitating" and "non-precipitating" grid points, KLD is reduced during the assimilation step. Experiments with longer windows show more KLD growth during the forecast as expected, but also a larger reduction at the analysis step, which is not as effective as the more frequent updates in reducing the analysis KLD. As noted before, the specific humidity over the "non-precipitating" grid points behaves differently, and KLD increases during the assimilation step for almost all times and experiments, leading to larger KLD at shorter assimilation windows (Figs. 6b, f). In this area mostly "non-precipitating" observations are assimilated to suppress spurious clouds. Interestingly in the "non-precipitating" grid points 5MIN-4D is the experiment providing the lowest KLD for all variables (Figs. 7b, d and f). This result suggests the potential benefits of treating "non-precipitating" observations differently."

[Figure]

Figure 7: Sawtooth time-series of the KLD (10$^{-2}$) of the analysis and first guess over the precipitating (<0dBZ, a,c,e) and non-precipitating (>30dBZ, b,d,f) grid points for temperature (a,b), specific humidity (c,d) and vertical velocity (e,f) and for the 5MIN (red), 5MIN-4D (blue), 2MIN (green), 1MIN (magenta), 1MIN-4D (black) and 30SEC (cyan) experiments.

6. Line 67: "… Climate Forecast System Reanalysis Saha et al. (2010)" à "… Climate Forecast System Reanalysis (Saha et al., 2010)"

We change this following the reviewer's comment.

7. Line 99 and Eq.(1): "where P(x) and Q(x) are two . . . " à should be "where p(x) and q(x) are two . . ."?

We agree with the reviewer and change the sentence accordingly. The new version of the sentence is on Line 114 of the new version of the manuscript: "where p(x) and q(x) are the probability density functions (PDFs) of P and Q, respectively."

8. Lines 140–141: Wrong use of "so that".

Following the reviewer's suggestion, we change the sentences in the following way (Line 185): "However, KLD in 5MIN-4D is larger than that in 30-SEC or 1MIN-4D, **indicating that** DA frequency is equally important."

9. Lines 144: "raining" and "non-raining" grid points sound better than "rain" and "non-rain" grid points to me.

Following the reviewer's suggestion we change the definition to "precipitating" and "non-precipitating" to avoid possible confusion with different types of precipitation.

---

## Author Response (AR2)

Reviewer #1

In the revised manuscript of "Reduced non-Gaussianity by 30-second rapid update in convective-scale numerical weather prediction", the authors have well addressed the previous problems and have included detailed information of the presented experiments, discussing the downside of their experiment results and objectively pointing out the non-inevitability of the causal link between reducing KLD and improving the EnKF performance. Overall, I agree this manuscript to be considered as a publication of NPG.

We would like to thank the reviewer #1 for all the comments that helped to improve the manuscript.

Reviewer #2

Review of "Reduced non-Gaussianity by 30-second rapid update in convective-scale numerical weather prediction" by Ruiz J., et al..
The authors made reasonable revisions in response to the comments from both reviewers. The revisions improved the manuscript. However I strongly recommend adding subheadings to the result section to make it easier to read and understand.

Once this minor revision is carried out I recommend the manuscript be accepted for publication in NPG.

We would like to thank the reviewer #2 for all the comments that helped to improve the manuscript.

We agree with the reviewer comment about adding subtitles to the result section. We split the result section into 3 subsections to improve the clarity of the discussion.